# Characterization of Beta-Lactamase and Fluoroquinolone Resistance Determinants in *Escherichia coli*, *Klebsiella pneumoniae,* and *Pseudomonas aeruginosa* Isolates from a Tertiary Hospital in Yola, Nigeria

**DOI:** 10.3390/tropicalmed8110500

**Published:** 2023-11-16

**Authors:** Diane E. Kawa, Isabella A. Tickler, Fred C. Tenover, Shuwaram A. Shettima

**Affiliations:** 1Department of Medical and Scientific Affairs, Cepheid, Sunnyvale, CA 94089, USA; 2Department of Medical and Scientific Affairs, Cepheid, 20090 Milan, Italy; isabella.tickler@cepheid.com; 3College of Arts and Sciences, University of Dayton, Dayton, OH 45469, USA; ftenover1@udayton.edu; 4Department of Medical Microbiology, Parasitology and Immunology, Modibbo Adama University Teaching Hospital, Yola 640001, Adamawa State, Nigeria; shuwy76@gmail.com

**Keywords:** beta-lactamase, carbapenem resistance, carbapenemase, fluoroquinolone resistance, Yola, Nigeria

## Abstract

Infections due to antimicrobial resistant gram-negative bacteria cause significant morbidity and mortality in sub-Saharan Africa. To elucidate the molecular epidemiology of antimicrobial resistance in gram-negative bacteria, we characterized beta-lactam and fluoroquinolone resistance determinants in *Escherichia coli*, *Klebsiella pneumoniae,* and *Pseudomonas aeruginosa* isolates collected from November 2017 to February 2018 (Period 1) and October 2021 to January 2022 (Period 2) in a tertiary medical center in north-eastern Nigeria. Whole genome sequencing (WGS) was used to identify sequence types and resistance determinants in 52 non-duplicate, phenotypically resistant isolates. Antimicrobial susceptibility was determined using broth microdilution and modified Kirby–Bauer disk diffusion methods. Twenty sequence types (STs) were identified among isolates from both periods using WGS, with increased strain diversity observed in Period 2. Common ESBL genes identified included *bla*_CTX-M_, *bla*_SHV,_ and *bla*_TEM_ in both *E. coli* and *K. pneumoniae*. Notably, 50% of the *E. coli* in Period 2 harbored either *bla*_CTX-M-15_ or *bla*_CTX-M-1 4_ and phenotypically produced ESBLs. The *bla*_NDM-7_ and *bla*_VIM-5_ metallo-beta-lactamase genes were dominant in *E. coli* and *P. aeruginosa* in Period 1, but in Period 2, only *K. pneumoniae* contained *bla*_NDM-7_, while *bla*_NDM-1_ was predominant in *P. aeruginosa*. The overall rate of fluoroquinolone resistance was 77% in Period 1 but decreased to 47.8% in Period 2. Various plasmid-mediated quinolone resistance (PMQR) genes were identified in both periods, including *aac(6*′*)-Ib-cr*, *oqxA/oqxB*, *qnrA1*, *qnrB1*, *qnrB6*, *qnrB18, qnrVC1*, as well as mutations in the chromosomal *gyrA*, *parC* and *parE* genes. One *E. coli* isolate in Period 2, which was phenotypically multidrug resistant, had ESBL *bla*_CTX-M-15,_ the serine carbapenemase, *bla*_OXA-181_ and mutations in the *gyrA* gene. The co-existence of beta-lactam and fluoroquinolone resistance markers observed in this study is consistent with widespread use of these antimicrobial agents in Nigeria. The presence of multidrug resistant isolates is concerning and highlights the importance of continued surveillance to support antimicrobial stewardship programs and curb the spread of antimicrobial resistance.

## 1. Introduction

The development and spread of antimicrobial resistance (AMR) among bacterial isolates continue to pose therapeutic challenges globally. In 2019, an estimated 4.95 million deaths were associated with AMR, with 1.27 million attributable deaths [1]. The highest burden of AMR is in western sub-Saharan Africa [1,2], where it is driven by inappropriate use (misuse and overuse) of antimicrobial agents [2,3]. 

Gram-negative bacteria, such as *Escherichia coli*, *Klebsiella pneumoniae*, *Enterobacter* spp., *Pseudomonas aeruginosa,* and *Acinetobacter baumannii*, are leading causes of antimicrobial-resistant infections [1] with resistance to fluoroquinolones and β-lactam drugs being widely reported [1,2]. Treatment of infections is often problematic due to the high level of intrinsic and acquired antimicrobial resistance genes. Plasmid-mediated transfer of genes encoding extended-spectrum beta-lactamases (ESBLs) and carbapenemases are increasingly reported in Africa and Asia [4,5,6,7,8]. Fluoroquinolone resistance is primarily driven by mutations in the quinolone resistance-determining regions (QRDR) of the DNA gyrase (*gyr*) and topoisomerase IV (*par*) encoding genes, overexpression of quinolone efflux pumps, and porin inhibition [9,10]. However, the emergence of plasmid-mediated quinolone resistance (PMQR) is particularly concerning and increasingly reported in the Enterobacterales [11,12].

Since the introduction of fluoroquinolones to Nigeria in 2004, there has been intense use of this class of broad-spectrum antibiotics both in and out of the hospital setting. Common indicators for use include urinary tract infections (UTIs), sepsis, enteric fever, acute bacterial gastroenteritis, acute otitis media, pelvic inflammatory disease, and perioperative antimicrobial prophylaxis. Consequently, over the last decade, there have been several reports of fluoroquinolone-resistant bacteria from human and animal sources [13,14,15,16]. Beta-lactam antibiotics are also widely used to treat UTIs caused by Enterobacterales and Pseudomonaceae [6] and reports of multidrug resistance associated with CTX-M-type ESBL-producing *E. coli* and *K. pneumoniae* have also increased [17]. 

At our hospital in Yola, Nigeria, the 2019 cumulative antimicrobial susceptibility test data (i.e., antibiogram), which guides empiric therapy, revealed that the percent susceptibility among gram-negative bacteria commonly isolated from clinical samples, was 2–64% for the beta-lactam drugs including meropenem, and 6–16% for ciprofloxacin, respectively (unpublished data). This prompted widespread clinician education about the appropriate use of these antimicrobial agents, and an improved percent susceptibility for the beta-lactam agents (22–76%) and for ciprofloxacin (28–59%) was observed in the 2021 antibiogram (unpublished data). Until recently, there was little information about the underlying mechanisms for antimicrobial resistance among gram-negative isolates at our institution and their prevalence. Using whole genome sequencing (WGS), we determined that the most common carbapenem-resistance mechanisms in gram-negative bacterial isolates were mediated by Class B metallo-beta-lactamases, including the New Delhi metallo-beta-lactamase *bla*_NDM-*7*_ and *bla*_NDM-1_ genes, the Verona integron-encoded metallo-beta-lactamase, *bla*_VIM-5,_ and Class D beta-lactamase *bla*_OXA-181_ [18,19]. This information was used to supplement the new 2021 antibiogram and further educate clinicians about antibiotic use. 

The goal of this study was to compare the molecular epidemiology and emergence of beta-lactam and fluoroquinolone resistance determinants in *E. coli*, *K. pneumoniae,* and *P. aeruginosa* isolates, the most common gram-negative isolates at our hospital, across two time periods (November 2017 to February 2018 and from October 2021 to January 2022). 

## 2. Materials and Methods

### 2.1. Bacterial Isolates

This study utilized non-duplicate *E. coli*, *K. pneumoniae,* and *P. aeruginosa* isolates from clinical specimens in the Medical Microbiology Laboratory at the Modibbo Adama University Teaching Hospital in Yola, Nigeria. A total of four *E. coli*, one *K. pneumoniae*, and 8 *P. aeruginosa* isolates collected from November 2017 to February 2018 (Period 1), previously characterized as carbapenem non-susceptible [18], and 19 *E. coli*, 12 *K. pneumoniae*, and 8 *P. aeruginosa* isolates collected between October 2021 to January 2022 (Period 2) were evaluated. The isolates were obtained from various clinical specimens; the most common source was urine (Table 1). The gram-negative isolates were identified to the species level using manual biochemical methods [20]. Susceptibility to antimicrobial agents was determined using the modified Kirby–Bauer disk diffusion method on Mueller–Hinton agar as described by the Clinical and Laboratory Standards Institute (CLSI) document M02-A13 [21]. The disk diffusion results were interpreted according to CLSI recommendations M100-S32 [22]. Gram-negative isolates that were non-susceptible/resistant to beta-lactam antimicrobial agents and fluoroquinolones were stored in physiological saline containing 25% glycerol in cryovials at −20 °C and shipped to a central laboratory for further analysis (described in Section 2.2). Ethical approval to collect bacterial isolates was obtained from the Federal Medical Centre, Yola Health Research Ethics Committee (HREC). 

### 2.2. Bacterial Identification and Antimicrobial Susceptibility Testing 

Identification of the bacterial isolates was performed at a central laboratory using MALDI-TOF MS (Bruker Daltonics GmbH, Bremen, Germany) according to the manufacturer’s instructions. Antimicrobial susceptibility testing was conducted using the Neg MIC 56 panel on the MicroScan WalkAway 40 SI Plus system (Beckman Coulter, Inc., West Sacramento, CA, USA) as described by the manufacturer, and MIC results were interpreted according to CLSI recommendations [22]. The antimicrobial agents were aztreonam (ATM), ceftazidime (CAZ), cefepime (FEP), cefotaxime (CTX), ceftriaxone (CRO), cefiderocol (FDC), piperacillin-tazobactam (TZP), ceftolozane-tazobactam (C/T), ceftazidime-avibactam (CZA), meropenem-avibactam (MVB), ertapenem (ETP), imipenem (IPM), meropenem (MEM), ciprofloxacin (CIP), levofloxacin (LVX), moxifloxacin (MXF). The concentrations of the antimicrobial agents included in the Neg MIC56 panel can be found at https://www.beckmancoulter.com/en/products/microbiology/-/media/63adeb88ba294285874b037c2ad875e4.ashx (accessed on 9 November 2023). Quality control organisms included *P. aeruginosa* ATCC 27853, *E. coli* ATCC 25922 and ATCC 35218, and *K. pneumoniae* ATCC 700603 and ATCC BAA-1705. The bacterial isolates were also tested for susceptibility to 12 antimicrobial agents using the modified Kirby–Bauer disk diffusion method on Mueller–Hinton agar (Hardy Diagnostics, Santa Maria, CA, USA) as described in CLSI M02-A13 [21]. Antimicrobial agents tested with disk diffusion included ATM (30 μg), CAZ (30 μg; with and without clavulanic acid), FEP (30 μg), CTX (30 μg; with and without clavulanic acid), CRO (30 μg), FDC (30 μg), CZA (30/20 μg), ETP (10 μg), IPM (10 μg), and MEM (30 μg); Zone diameters and ESBL detection were interpreted using the criteria described in CLSI M100-S32 [22]. Quality control organisms included *P. aeruginosa* ATCC 27853, *E. coli* ATCC 25922 and ATCC 35218, and *K. pneumoniae* ATCC 700603 and ATCC BAA-1705. 

### 2.3. Phenotypic Detection of Carbapenemase Production

The modified carbapenem inactivation method (mCIM) was used in conjunction with the EDTA-modified carbapenem inactivation method (eCIM) to determine carbapenemase production and to differentiate metallo-beta-lactamases from serine carbapenemases, according to CLSI guidelines [22].

### 2.4. Whole Genome Sequencing 

Genomic DNA was extracted from pure cultures of organisms grown overnight on blood agar plates (Hardy Diagnostics) using the Qiagen DNeasy blood and tissue kit on the Qiacube (Qiagen, Valencia, CA, USA). Genomic libraries were prepared for each isolate using the Illumina DNA Prep Kit (Illumina, San Diego, CA USA), and sequencing was performed on the Illumina Miseq system using Miseq Reagent Kit v2 (Illumina, San Diego, CA) according to the manufacturer’s instructions. De novo assemblies, multi-locus sequence typing (MLST), detection of acquired antimicrobial resistance genes, and detection of point mutations were performed with the CLC Genomics Workbench version 22.0.2 and CLC Microbial Genomics Module version 22.1.1 (QIAGEN Bioinformatics, Aarhus, Denmark). All nucleic acid sequence data from this study have been deposited in the NCBI BioProject database (https://www.ncbi.nlm.nih.gov/bioproject/, accessed on 9 November 2023) with links to BioProject Accession # PRJNA701275 (2017–2018 isolates) and # PRJNA962793 (2021–2022).

## 3. Results

### 3.1. Antimicrobial Resistance Profiles

The beta-lactam and fluoroquinolone resistance profiles of the *E. coli*, *K. pneumoniae,* and *P. aeruginosa* isolates from Period 1, based on antimicrobial suscpetibility testing performed at the central laboratory, are shown in Figure 1. The overall prevalence of antimicrobial resistance among the 13 isolates was 76.9%. The highest rate was in *E. coli*, which demonstarted 100% resistance to all classes of antimicrobial drugs tested. The only *Klebsiella pneumoniae* isolate 16047, was characterized as phenotypically positive for ESBL production (Appendix A). Ten isolates were non-susceptible to one or more carbapenems and of these, nine isolates (4 *E. coli* and 5 *Pseudomonas aeruginosa*) produced a metallo-beta-lactamase (Appendix A).

In contrast to Period 1, the overall resistance rate among the 39 isolates from Period 2 was 53.8% for beta-lactams, 33.8% for carbapenems, and 56.4% for fluoroquinolones (Figure 2). The highest rate was for *P. aeruginosa*, with 100% resistance to the carbapenems. A total of 12 isolates produced ESBLs. Of the 12 isolates that were non-susceptible to the carbapenems, three *Klebsiella pneumoniae* and six *Pseudomonas aeruginosa* produced a metallo-beta-lactamase (Appendix A). Notably, one *E. coli* isolate (17781) produced an ESBL and a serine carbapenemase as determined by mCIM/eCIM (Appendix A). There was 100% concordance in antimicrobiaol susceptibility test results between the broth microdilution and disk diffusion methods for the isolates.

### 3.2. Strain Types and Resistance Determinants among E. coli Isolates

The multi-locus sequence type (MLST) and resistance determinants for the *E. coli* isolates from Period 1 are presented in Table 2. All four *E. coli* ST692 isolates harbored ESBL *bla*_CTX-M-15_ gene and metallo-beta-lactamase *bla*_NDM-7_ gene. These genetic profiles were consistent with the beta-lactam and carbapenem resistance phenotypes observed (Table 2). The isolates also had the PMQR *aac*(*6*′)*Ib-cr* gene and mutations in the *parC* and *parE* genes, respectively. Three isolates had mutations in *gyrA* consistent with the fluoroquinolone-resistant phenotype of the isolates (Appendix A). 

In Period 2, there were multiple sequence types for *E. coli,* with ST2 being dominant (Table 3). ESBL-encoding genes bla_CTX-M-15_ and *bla*_CTX-M-14_ were identified in 57.9% of the isolates and all (100%) were phenotypically ESBL producers (Appendix A). The distribution of PMQR genes and mutations in the *gyrA, parC* and *parE* genes also varied. Five *E. coli* isolates (26.3%) had one or more PMQR genes and 14 (73.6%) had at least one mutation in *gyrA*, *parC,* or *parE.* Of the 12 isolates that were fluoroquinolone resistant (Appendix A), 11 (91.6%) harbored two mutations in gyrA, suggesting this was the main mechanism of resistance in these isolates. One *E. coli* isolate (17781; ST692) harbored multiple resistance markers including *bla*_CTX-M-15,_
*bla*_OXA-181_, and mutations in *gyrA* (Table 3). Notably, this *E. coli* isolate produced an ESBL, a serine carbapenemase, and was phenoptypically non-susceptible to ertapenem and the fluoroquinolones (Appendix A).

### 3.3. Strain Types and Resistance Determinants in K. pneumoniae Isolates

The sole *K. pneumoniae* isolate (16047; ST147) from Period 1, had *bla*_CTX-M-15_, four PMQR genes, and mutations in the *gyrA* gene (Table 2), was an ESBL-producer and was phenotypically non-susceptible to the carbapenems (Appendix A). In contrast, the 12 *K. pneumoniae* isolates from Period 2 represented seven distinct sequence types (Table 3). Almost all the *K. pneumoniae* isolates (83.3%) from this period had at least one beta-lactamase gene present. Although only one *K. pneumoniae* isolate (17785; ST45) had *bla*_CTX-M-15_ and was an ESBL producer, three isolates (25%) harbored *bla*_NDM-7_, produced a metallo-beta-lactamase and were phenotypically non-susceptible to carbapenems (Appendix A). The prevalence of PMQR genes *oqxA and oqxB* among *K. pneumoniae* in Period 2 was 100%. No QRDR mutations were detected in these isolates; however, all were resistant to the fluoroquinolones (Appendix A), suggesting that the PMQR determinants were associated with the resistant phenotype observed in Period 2.

### 3.4. Strain Types and Resistance Determinants in P. aeruginosa Isolates 

The *P. aeruginosa* isolates from Period 1 represent an array of sequence types (Table 2). The overall prevalence of carbapenemase genes among these isolates was 62%. Four isolates (50%) harbored *bla*_VIM-5,_ one had *bla*_NDM-1,_ and all five were phenotypically non-susceptible to the carbapenems via production of metallo-beta-lactamases (Appendix A). Interestingly, these same five *P. aeruginosa* isolates also had *qnrVC1*, identical mutations in the *gyrA* and *parC* genes (Table 2), and were all resistant to fluoroquinolones (Appendix A). It is unclear which fluoroquinole-resistance deteminants were directly responsible for the resistant phenotype observed. 

The majority of the *P. aeruginosa* isolates (75%) from Period 2 were ST773 and had identical resistance determinants including, *bla*_NDM-1_, *qnrVC1* genes, and mutations in *gyrA* and *parC* (Table 3). Not surprisingly, these six isolates showed a corresponding phenotype for carbapenem non-susceptibility and fluoroquinolone resistance (Appendix A). Although the remaining two *P. aeruginosa* isolates were also carbapenem-resistant, they lacked a known carbapenemase gene. 

## 4. Discussion

As part of our antimicrobial resistance surveillance efforts, we characterized beta-lactam and fluoroquinolone resistance determinants in *E. coli*, *K. pneumoniae,* and *P. aeruginosa* isolates collected within two time periods from November 2017 to February 2018 and from October 2021 to January 2022, at the Modibbo Adama University Teaching Hospital in Yola, which is located in the north-eastern region of Nigeria. The findings of this study will be used to supplement our annual antibiograms that guide clinician decisions for empiric treatment for gram-negative bacterial infections, especially UTIs. 

There was considerable diversity in sequence types among the gram-negative isolates across the two study periods. Notably, *E. coli* ST131, which is a highly virulent strain associated with multidrug resistance [23] and has recently been reported in Central Nigeria [24], was not identified at our institution. The high prevalence of the *bla*_CTX-M-15_ and *bla*_CTX-M-14_ ESBLs among the *E. coli* isolates in our study is consistent with the regional and global distribution of these resistance mechanisms. For instance, in a study characterizing multidrug resistant *E. coli* isolates in Abuja, Nigeria (Central region), Medugu et al., reported that *bla*_CTX-M-15_ and expression of a ESBL phenotype were detected in 70.1% and 50% of the isolates, respectively [24]. Similarly, in a study characterizing multidrug resistant uropathogenic Enterobacterales, and Pseudomonaceae at their institution in south-west Nigeria, Ogbolu et al., found that *bla*_CTX-M-15_ was the dominant CTX-M gene (83.3%) and positivity for *bla*_CTX-M-14_ was 33.3%. Some *E. coli* and *K. pneumoniae* isolates carried both genes and although CTX-M genes are not typically found in *P. aeruginosa*, two isolates were also shown to carry *bla*_CTX-M-15_ [17]. In a recent study in the United States of America, ESBL genes were identified in 66.2% of gram-negative bacteria isolated from urine and blood specimens, with *bla*_CTX-M-15_ being the most common [25].

We observed a shift in the dominant metallo-beta-lactamase from *bla*_NDM-7_ and *bla*_VIM-5_ in Period 1 to primarily the *bla*_NDM-1_ gene in Period 2, although the occurrence of this carbapenemase in several ST 773 *P. aeruginosa* isolates suggests it was possibly related to clonal expansion. A mutation in the *oprD* gene was identified in one carbapenem-non-susceptible *P. aeruginosa* that lacked a carbapenemase gene; however, no mutations associated with porin alterations or overexpression of efflux pumps that could lead to a resistance phenotype [9], were observed in the other isolate. Various carbapenemase genes, including *bla*_NDM-7_, *bla*_NDM-1_ and *bla*_OXA-181_ have been identified in Nigeria [26] and other parts of Africa [7]. 

The high rate of fluoroquinolone resistance among *E. coli*, *K. pneumoniae,* and *P. aeruginosa* isolates in Period 1 (77%) and Period 2 (47.8%) was not surprising, given the widespread use of fluoroquinolones in Nigeria. Consistent with our findings, a study at a tertiary hospital in southern Nigeria revealed that 93.3% of *E. coli* isolates harboured at least one fluoroquinolone resistance gene [27]. Similar patterns of high levels of fluoroquinolone resistance have been reported in other parts of sub-Saharan Africa including South Africa and Kenya [10,28,29,30,31]. In our study, fluoroquinolone resistance in *E.coli* and *P. aeruginosa* was primarily conferred by double or triple mutations in the *gyrA* gene. While high-level fluoroquinolone resistance is not typically associated with PMQR genes, their presence in some isolates suggests a likely role in decreasing fluoroquinolone susceptibility. Co-existence of PMQR determinants and mutations in the QRDR regions of the *gyrA* and *parC* housekeeping genes has been documented as mediating higher levels of resistance [32,33]. In their study in Poland, Pierkaska et al., suggest that PMQR genes may contribute to promoting the mutations of the QRDR leading to increased fluoroquinolone non-susceptibility [32]. 

In our study, several isolates co-harbored *bla*_CTX-M-15_ and *bla*_CTX-M-14_ ESBL genes, and *bla*_NDM-7_ and *bla*_NDM-1_ carbapenemase genes together with various combinations of PMQR genes. The possibility that ESBL and PMQR genes may co-exist on the same plasmid is worrisome because of the potential for rapid horizontal transfer of resistance between bacterial strains. One *E. coli* isolate (17781; ST692) from the second study period carried the *bla*_OXA-181_ carbapenemase gene and exhibited a serine carbapenemase phenotype. This isolate, which was recovered from a urine specimen, is of particular concern because it co-harbored *bla*_CTXM-15_, a Class D AmpC beta-lactamase gene, *bla*_CMY-2_, the PMQR *qnrS1* gene, and had multiple mutations in the *gyrA*, *parC,* and *parE* genes. Not surprisingly, the isolate was non-susceptible to all fluoroquinolone and beta-lactam antimicrobials except cefiderocol. The potential spread of this multidrug resistant *E. coli* strain poses a significant threat to the treatment of urinary tract infections, which are common in our region.

This study had some limitations. First, the analysis was performed with *E. coli*, *K. pneumoniae* and *P. aeruginosa* isolates, and does not represent all the gram-negative bacterial organisms isolated at our institution that may contain fluoroquinolone and beta-lactam resistance markers. Secondly, the *E. coli*, *K. pneumoniae,* and *P. aeruginosa* isolates from Period 1 were a subset of carbapenem- non-susceptible isolates from our hospital and this may have introduced a pre-selection bias in favour of isolates with a high level of resistance to carbapenems and other antimicrobial drugs. However, several isolates in Period 2, which had much broader selection criteria, also had multiple resistant determinants and demonstrated a resistant or non-susceptible phenotype for the beta-lactam and fluoroquinolone antimicrobials.

In conclusion, our study showed significant diversity in the sequence types and co-existence of beta-lactam and fluoroquinolone resistance determinants in *E. coli*, *K. pneumoniae* and *P. aeruginosa* isolates at our institution. Due to the high mortality and morbidity associated with antimicrobial resistance in gram-negative bacteria, our findings underscore the importance of continued molecular surveillance for existing and emerging resistant organisms that can inform therapeutic decisions and antimicrobial stewardship programs in healthcare settings, especially in sub-Saharan Africa.

## Figures and Tables

**Figure 1 tropicalmed-08-00500-f001:**
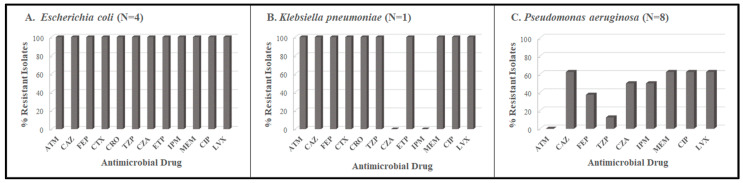
Beta-lactam and fluoroquinolone resistance profiles for *E. coli*, *K. pneumoniae,* and *P. aeruginosa* isolates collected during Period 1. ATM: aztreonam; CAZ: ceftazidime; FEP: cefepime; CTX: cefotaxime; CRO: ceftriaxone; TZP: piperacillin/tazobactam; CZA: ceftazidime/avibactam; ETP: ertapenem; IPM: imipenem; MEM: meropenem; CIP: ciprofloxacin; LVX: levofloxacin.

**Figure 2 tropicalmed-08-00500-f002:**
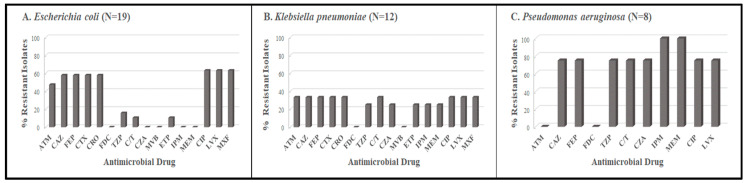
Beta-lactam and fluoroquinolone resistance profiles for *E. coli*, *K. pneumoniae* and *P. aeruginosa* isolates collected during Period 2. ATM: aztreonam; CAZ: ceftazidime; FEP: cefepime; CTX: cefotaxime; CRO: ceftriaxone; FDC: cefiderocol; TZP: piperacillin/tazobactam; C/T: ceftolozane-tazobactam; CZA: ceftazidime/avibactam; MVP: meropenem-avibactam; ETP: ertapenem; IPM: imipenem; MEM: meropenem; CIP: ciprofloxacin; LVX: levofloxacin.

**Table 1 tropicalmed-08-00500-t001:** Select antimicrobial-resistant *Escherichia coli*, *Klebsiella pneumoniae* and *Pseudomonas aeruginosa* isolates recovered from clinical specimens from November 2017 to February 2018 (Period 1) and October 2021 to January 2022 (Period 2).

Collection Period	Organism	Blood Culture	Ear Swab	Endocervical Swab	Eye Swab	Stool	Vaginal Swab	Sputum	Urethral Swab	Urine	Wound	Total
Period 1 November 2017–February 2018	*Escherichia coli*		1							3		4
	*Klebsiella pneumoniae*									1		1
	*Pseudomonas aeruginosa*					1		2		3	2	8
Period 2 October 2021–January 2022	*Escherichia coli*						3			12	4	19
	*Klebsiella pneumoniae*	1		1				4	1	3	2	12
	*Pseudomonas aeruginosa*		2		1			1		3	1	8
	Total	1	3	1	1	1	3	7	1	25	9	52

**Table 2 tropicalmed-08-00500-t002:** Beta-lactamase and fluoroquinolone resistance determinants and strain types among *E. coli, K. pneumoniae,* and *P. aeruginosa* isolates collected during Period 1.

ID	Organism	Source	MLST	Beta-Lactamase Genes	Fluoroquinolone PMQR Genes	Fluoroquinolone QRDR Mutations		
						*gyrA*	*parC*	*parE*
15949	*Escherichia coli*	ear swab	ST692	*bla*_CMY-59_, *bla*_CTX-M-15_, *bla*_NDM-7_, *bla*_OXA-1_, *bla*_TEM-1_, *ampH*	*aac*(*6*′)*Ib-cr*	S83L, D87N	S80I	S458A
16020	*Escherichia coli*	urine	ST692	*bla*_CMY-59_, *bla*_CTX-M-15_, *bla*_NDM-7_, *bla*_OXA-10_, *bla*_OXA-140_, *bla*_TEM-1_, *ampH*, *ampC*	*aac*(*6*′)*Ib-cr*	WT	S80I	S458A
16029	*Escherichia coli*	urine	ST692	*bla*_CMY-59_, *bla*_CTX-M-15_, *bla*_NDM-7_, *bla*_TEM-1_, *ampH*, *ampC*	*aac*(*6*′)*Ib-cr*	S83L, D87N	S80I	S458A
16032	*Escherichia coli*	urine	ST692	*bla*_CTX-M-15_, *bla*_NDM-7_, *bla*_OXA-1_, *bla*_CMY-59_, *bla*_TEM-1_, *ampH*, *ampC*	*aac*(*6*′)*Ib-cr*	S83L, D87N	S80I	S458A
16047	*Klebsiella pneumoniae*	urine	ST147	*bla*_CTX-M-15_, *bla*_OXA-1_, *bla*_SHV-187_, *K. pneumoniae* OmpK37, *E. coli ampH*	*aac*(*6*′)*Ib-cr oqxA, oqxB qnrA1, qnrB1*	S83Y, D87A	WT	WT
15958	*Pseudomonas aeruginosa*	sputum	ST2935	*bla*_OXA-50_, *bla*_PDC-10_	None	S912del, E913del	F254V, S331T, A346Q	ND
15964	*Pseudomonas aeruginosa*	wound	ST1203	*bla*_GES-9_, *bla*_OXA-21_, *bla*_OXA-50_, *bla*_PDC-1_, *bla*_VIM-5_	*aac*(*6*′)*Ib-cr, qnrVC1*	T83I, S912del, E913del	S87L, F254V, A346Q	ND
15965	*Pseudomonas aeruginosa*	urine	ST773	*bla*_NDM-1_, *bla*_OXA-50_, *bla*_PDC-1_	*qnrVC1*	T83I	S87L, F254V, A346Q	ND
15966	*Pseudomonas aeruginosa*	urine	ST1203	*bla*_GES-9_, *bla*_OXA-21-like_, *bla*_OXA-50_, *bla*_PDC-1_, *bla*_VIM-5_	*aac*(*6*′)*Ib-cr, qnrVC1*	T83I, S912del, E913del	S87L, F254V, A346Q	ND
15986	*Pseudomonas aeruginosa*	wound	ST654	*bla*_OXA-10_, *bla*_OXA-50_, *bla*_PDC-3_, *bla*_VIM-5_	*aac*(*6*′)*Ib-cr, qnrVC1*	T83I, S912del, E913del	S87L, F254V, A346Q	ND
16014	*Pseudomonas aeruginosa*	stool	ST244	*bla*_OXA-486_, *bla*_PDC-1_	None	WT	F254V, A346Q	ND
16018	*Pseudomonas aeruginosa*	urine	ST654	*bla*_OXA-10_, *bla*_OXA-50_, *bla*_PDC-3_, *bla*_TEM-1_, *bla*_VIM-5_, *bla*_SCO-1_	*aac*(*6*′)*Ib-cr, qnrVC1*	T83I, S912del, E913del	S87L, F254V, A346Q	WT
16048	*Pseudomonas aeruginosa*	sputum	ST1555	*bla*_OXA-50_, *bla*_PDC-10_	None	S912del, E913del	F254V, S331T, A346Q	ND

MLST, multi-locus sequence type; PMQR, plasmid-mediated quinolone resistance; QRDR, Quinolone Resistance Determinant Region; WT, wild type; ND, not determined.

**Table 3 tropicalmed-08-00500-t003:** Beta-lactamase and fluoroquinolone resistance determinants and strain types among *E. coli, K. pneumoniae,* and *P. aeruginosa* isolates collected during Period 2.

ID	Organism	Source	MLST	Beta-Lactamase Genes	Fluoroquinolone PMQR Genes	Fluoroquinolone QRDR Mutations		
						*gyrA*	*parC*	*parE*
17757	*Escherichia coli*	urine	Ambiguous (ST506, ST566)	*bla*_CTX-M-14_, *bla_TEM-1B_*	None	S83L, D87G	S80I	S458A, I529L
17758	*Escherichia coli*	VS	Ambiguous (ST27, ST129)	*bla* _TEM-1B_	None	S83L	WT	WT
17762	*Escherichia coli*	urine	Ambiguous (ST27, ST129)	*bla* _TEM-1B_	None	S83L	WT	WT
17771	*Escherichia coli*	urine	ST2	*bla* _TEM-1B_	None	WT	WT	WT
17772	*Escherichia coli*	urine	ST2	*bla*_CTX-M-15_, *bla*_OXA-1_, *bla*_TEM-1B_	*aac*(*6*′)*-Ib-cr*	S83L, D87N	S80I	S458A
17773	*Escherichia coli*	VS	ST83	None	*qnrB7*	WT	WT	WT
17775	*Escherichia coli*	urine	Inconclusive (ST721, ST662, ST472)	*bla*_CTX-M-15_, *bla*_OXA-_1, *bla*_TEM-1B_	*aac*(*6*′)*-Ib-cr*	S83L, D87N	S80I	S458A
17776	*Escherichia coli*	urine	Inconclusive (ST466, ST210, ST132)	*bla* _TEM-1B_	None	S83L, D87N	S80I	S458A
17781	*Escherichia coli*	urine	ST692	*bla*_CMY-2_, *bla*_CTX-M-15_, *bla*_OXA-181_, *bla*_TEM-1B_	*qnrS1*	S83L, D87N	S80I	S458A
17782	*Escherichia coli*	urine	ST471	*bla*_CTX-M-15_, *bla*_OXA-1_	None	S83L, D87N	S80I	S458A
17786	*Escherichia coli*	wound	ST132	*bla* _TEM-1B_	None	S83L, D87N	S80I	S458A
17789	*Escherichia coli*	urine	Inconclusive (ST500, ST437)	*bla* _TEM-1B_	*qnrS1*	WT	WT	WT
17792	*Escherichia coli*	wound	ST2632	*bla*_CTX-M-15_, *bla*_OXA-1_, *bla*_TEM-1B_	*aac*(*6*′)*-Ib-cr*	S83L, D87N	S80I	S458A
17795	*Escherichia coli*	urine	ST86	*bla*_CTX-M-15_, *bla*_TEM-1B_	*qnrS1*	S83L, D87N	S80I	
17796	*Escherichia coli*	urine	Ambiguous (ST566, ST506)	*bla*_CTX-M-14_, *bla*_TEM-1B_	None	S83L, D87G	S80I	S458A, I529L
17803	*Escherichia coli*	wound	ST2	*bla*_CTX-M-15_, *bla*_OXA-1_, *bla*_SHV-187_, *bla*_TEM-1B_	*aac*(*6*′)*-Ib-cr, oqxA*, *oqxB*	S83L, D87N	S80I	S458A
17804	*Escherichia coli*	wound	ST471	*bla* _CTX-M-15_	*aac*(*6*′)*-Ib-cr*	WT	WT	WT
17862	*Escherichia coli*	VS	Inconclusive (ST500, ST437)	*bla* _TEM-1B_	*qnrS1*	WT	WT	WT
17864	*Escherichia coli*	urine	ST58	*bla*_CTX-M-15_, *bla*_OXA-1_, *bla*_OXA-320/534_, *bla*_TEM-1B_	None	S83L, D87N	S80I	S458T
17759	*Klebsiella pneumoniae*	urine	ST340	*bla*_CTX-M-27_, *bla*_NDM-7_, *bla*_SHV-187_	*aac*(*6*′)*-Ib-cr, oqxA, oqxB, qnrB6*	WT	WT	WT
17765	*Klebsiella pneumoniae*	sputum	ST86	*bla*_OXA-1_, *bla*_SHV-187_	*aac*(*6*′)*-Ib-cr, oqxA, oqxB*	WT	WT	WT
17766	*Klebsiella pneumoniae*	sputum	ST20	*bla* _SHV-187_	*oqxA, oqxB*	WT	WT	WT
17767	*Klebsiella pneumoniae*	US	ST392	*bla* _SHV-11_	*aac*(*6*′)*-Ib-cr, oqxA, oqxB qnrB6*	WT	WT	WT
17768	*Klebsiella pneumoniae*	urine	ST340	*bla*_CTX-M-27_, *bla*_NDM-7_, *bla*_SHV-187_	*aac*(*6*′)*-Ib-cr, oqxA, oqxB, qnrB6*	WT	WT	WT
17770	*Klebsiella pneumoniae*	BC	ST86	*bla* _SHV-187_	*oqxA, oqxB*	WT	WT	WT
17785	*Klebsiella pneumoniae*	wound	ST45	*bla*_CTX-M-15_, *bla*_SHV-187_, *bla*_TEM-1B_	*oqxA, oqxB*	WT	WT	WT
17787	*Klebsiella pneumoniae*	sputum	ST2632	*bla* _SHV-93_	*oqxA, oqxB*	WT	WT	WT
17790	*Klebsiella pneumoniae*	wound	ST86	*bla* _SHV-187_	*oqxA, oqxB*	WT	WT	WT
17793	*Klebsiella pneumoniae*	sputum	ST661	*bla* _SHV-187_	*oqxA, oqxB*	WT	WT	WT
17861	*Klebsiella pneumoniae*	urine	ST86	*bla* _SHV-187_	*oqxA, oqxB*	WT	WT	WT
17863	*Klebsiella pneumoniae*	ES	ST340	*bla*_CTX-M-27_, *bla*_NDM-7_, *bla*_SHV-187_	*aac*(*6*′)*-Ib-cr, oqxA, oqxB, qnrB18, qnrB6*	WT	WT	WT
17760	*Pseudomonas aeruginosa*	sputum	ST274	*bla*_OXA-486_, *bla*_PDC-24_	None	Silent	F254V, A346Q	ND
17761	*Pseudomonas aeruginosa*	ear swab	Inconclusive (nearest ST244)	*bla*_OXA-847_, *bla*_PDC-423_	None	Silent	F254V, A346Q	ND
17763	*Pseudomonas aeruginosa*	urine	ST773	*bla*_NDM-1_, *bla*_OXA-395_, *bla*_PDC-385_	*qnrVC1*	T83I	S87L, F254V, A346Q	ND
17777	*Pseudomonas aeruginosa*	urine	ST773	*bla*_NDM-1_, *bla*_OXA-395_, *bla*_PDC-385_	*qnrVC1*	T83I	S87L, F254V, A346Q	ND
17778	*Pseudomonas aeruginosa*	ear swab	ST773	*bla*_NDM-1_, *bla*_OXA-395_, *bla*_PDC-385_	*qnrVC1*	T83I	S87L, F254V, A346Q	ND
17779	*Pseudomonas aeruginosa*	urine	ST773	*bla*_NDM-1_, *bla*_OXA-395_, *bla*_PDC-385_	*qnrVC1*	T83I	S87L, F254V, A346Q	ND
17791	*Pseudomonas aeruginosa*	eye swab	ST773	*bla*_NDM-1_, *bla*_OXA-395_, *bla*_PDC-385_	*qnrVC1*	T83I	S87L, F254V, A346Q	ND
17794	*Pseudomonas aeruginosa*	wound	ST773	*bla*_NDM-1_, *bla*_OXA-395_, *bla*_PDC-385_	*qnrVC1*	T83I	S87L, F254V, A346Q	ND

VS, vaginal swab; ES, endocervical swab; US, urethral swab; BC, blood culture; MLST, multi-locus sequence type; PMQR, plasmid-mediated quinolone resistance; QRDR, Quinolone Resistance Determinant Region; WT, wild type; ND, not determined; Silent, synonymous mutation with no change in amino acid sequence.

## Data Availability

Data are contained within the article and Appendix A.

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
