# Peer review of "Characterization of Beta-Lactamase and Fluoroquinolone Resistance Determinants in Escherichia coli, Klebsiella pneumoniae, and Pseudomonas aeruginosa Isolates from a Tertiary Hospital in Yola, Nigeria"

_tropicalmed, 2023, doi:10.3390/tropicalmed8110500_

Round 1
Reviewer 1 Report
Comments and Suggestions for Authors
Review of the manuscript tropicalmed-2667367
Title of the manuscript: Characterization of Beta-Lactamase and Fluoroquinolone Resistance Determinants in Escherichia coli, Klebsiella pneumoniae and Pseudomonas aeruginosa isolates from a Tertiary Hospital in Yola, Nigeria
Brief description of the work: Antimicrobial resistance was determined in E. coli, Klebsiella pneumoniae and Pseudomonas aeruginosa isolates from Medical Microbiology Laboratory at the Modibbo Adama University Teaching Hospital in Yola, Nigeria. These isolates were collected in the two time frames i.e. between November 2017 and February 2018 (Period 1) and between October 2021 to January 2022 (Period 2).Mark differences in sequence type was observed for all isolates and only few may have originated from the same clone as they were detected in both periods (such as P. aeruginosa ST773 or E. coli ST692). It seems that antibiotic use is a driven force in development of resistance to antibiotics in the hospital and that simultaneous occurrence of resistance to carbapeneme and fluoroquinolone is a trend for many years.
Specific comments:
Introduction
Lines 55-59: The goal of the present study was to characterize the distribution of beta-lactam and fluoroquinolone resistance determinants in E. coli, K. pneumoniae and P. aeruginosa isolates collected at a tertiary hospital in Yola, Nigeria from November 2017 to February 2018 and from October 2021 to January 2022.
The research goal is not clear. Are you conducting a brief epidemiology work on this occasion? Why it is important to compare two Periods in terms of AMR of bacteria isolates from the Hospital in Yola? Did previous research prompted advanced measures in Hospital hygiene practice to lower the occurrence of carbapenemase producing bacteria? Please explain in more detail the reasons for conducting this research.
Specific comments:
Material and methods
Lines 62-65: This study utilized a convenience sample of 19 non-duplicate E. coli, 12 K. pneumoniae and eight P. aeruginosa isolates from clinical specimens from the Medical Microbiology Laboratory at the Modibbo Adama University Teaching Hospital in Yola, Nigeria from 64 October 2021 to January 2022.
What does it mean “convenience sample”?
Lines 81-84: Antimicrobial susceptibility testing was conducted using the Neg MIC 56 panel on the MicroScan WalkAway 40 SI Plus system (Beckman Coulter, Inc., West Sacramento, CA, USA) as described by the manufacturer and MIC results were interpreted according to CLSI recommendations [16].
Please add “M100-S32” before reference number.
Please write more systematically material and methods section “2.1. Bacterial isolates”.
This can be achieved by mentioning how bacteria identification was done, how they were stored, what isolates were included in the study for Period 1 and Period 2 (Table 1) and after that explain how antimicrobial susceptibility was tested.
Therefore the best will be to move from the result section to material and methods the following sentences and Table 1, since -the origin of isolates- is not a result:
Lines 122-126: A total of four E. coli, one K. pneumoniae, and eight P. aeruginosa isolates collected from November 2017 to February 2018 (Period 1) and 19 E. coli, 12 K. pneumoniae, and eight P. aeruginosa isolates collected between October 2021 to January 2022 (Period 2) were evaluated (Table 1). The isolates were obtained from various clinical specimens; the most common source was urine (Table 1).
Lines 97-98: Zone diameters and ESBL detection were interpreted using the criteria described in CLSI M100 [16].
Please make corrections “CLSI M100-S32” instead of “CLSI M100”
Results
Lines 129-132: The four E. coli isolates were 100% resistant to all classes of drugs, including the cephems (ceftazidime, ceftazidime, cefepime, cefotaxime, ceftriaxone) and carbapenems, as well as the fluoroquinolones (Figure 1A).
Please delete “ceftazidime” once. Indicate if these isolates were ESBL producers.
Please add abbreviation of antibiotics, bellow the Fig 1 and 2.
Please do not retell data from Fig 1, Fig 2 and Tables 2 and 3. Rather compare the resistance rate in these periods. Then, describe how many isolates were characterized as ESBL and how many isolates produced acquired carbapenemase. Also indicate number of isolates with plasmid mediated resistance to FQ and number of isolates with the mutations on topoisomerase genes. This will provide a close summary of these two Graphs and Tables. Was the phenotype in agreement with genotype?
Please see bellow comments on Discussion.
Discussion
Most of the discussion needs to be in the result section. Are your results comparable to similar research? Tell us more about studies on resistant bacteria conducted in Nigeria. How is your work different or improved comparing to other research from your country? What is the situation in developed countries? Is resistance rate in clinical isolates as high as in other countries? How is your research going to help medical practitioners in Nigeria? Which antibiotics could be used for the treatment of patients?
Here are sentences from Discussion which, according to my opinion, need to be transferred to the results section:
Lines 262-265: “For instance, E. coli ST692 was dominant in Period 1 but there were over seven known STs and several ambiguous or inconclusive STs for E. coli during the second period. In contrast, there were seven different STs for P. aeruginosa in the first period of the study, while ST773 predominated in Period 2. Notably, no E. coli ST131 isolates were identified at our institution.”
Lines 269-272: “Our study revealed a high prevalence and co-existence of blaCTX-M, blaSHV, and blaTEM ESBL genes in E. coli and K. pneumoniae isolates from both periods. All ESBL-producing isolates harbored either blaCTX-M-15 or blaCTX-M-14, which were found only in E. coli; over 50% of the E. coli isolates in the second period were ESBL producers.”
Lines 276-279: “In contrast to Period 1 in which the metallo-beta-lactamases blaNDM-7 and blaVIM-5 were the most prevalent carbapenemase determinants in E. coli and P. aeruginosa, respectively, we observed that in Period 2, blaNDM-7 occurred mainly in K. pneumoniae while six P. aeruginosa isolates had the blaNDM-1 gene.”
Lines 288-293: “Although the distribution of the PMQR genes varied in our study, all fluoroquinolone-resistant E. coli isolates had mutations in the parC and parE genes. The fluoroquinolone-resistant K. pneumoniae isolates all had aac(6')-Ib-cr and oqxA/oqxB, and either qnrA1, qnrB1, qnrB6 or qnrB18 genes. Similarly, all fluoroquinolone-resistant P. aeruginosa isolates contained the qnrVC1 PMQR gene and three mutations in the parC gene”
I do not understand the text marked yellow. E. coli isolates from both periods (1 and 2) had double mutations on the gyrA gene. This will create significant (clinical) resistance to FQ antibiotic. Please rewrite this section and move it to the Results.
Lines 319-320: These organisms were not represented in the second period of this study and were therefore excluded from analysis.
Why are you mentioning Providencia rettgeri, Enterobacter cloacae and non-fermenters Stenotrophomonas maltophilia. There is no need for that since the focus of your work was E. coli, Klebsiella pneumonia and P. aeruginosa.
Line 330-331: Due to the high mortality and morbidity associated with AMR due to gram-negative bacteria…..
Please rewrite …“associated with antimicrobial resistance of Gram-negative bacteria….”
Opinion:
The research goal has to be clearly stated in the Introduction. Please improve Results section and Discussion. In result section comments should be more generalized since now it is difficult to follow up what are the major discoveries in this work. Please provide more comprehensive review of published works in respect to the results of your research.

Author Response
"Please see the attachment"

Reviewer 2 Report
Comments and Suggestions for Authors
Global comment
In this paper, the authors used 52 samples obtained from a Nigerian hospital and collected within two time periods, 13 isolates of Escherichia coli, Klebsiella pneumoniae and Pseudomonas aeruginosa were previously identified and characterized and 39 new isolates were analyzed belonging to the same bacterial species. Antimicrobial susceptibility testing was also performed, as well as sequencing, depositing the obtained sequences on the NCBI database.
This study allows us to monitor the antimicrobial resistance in human medicine by comparing two time periods, although with number of samples is low. In general, the manuscript is well-written with scientific quality. However, the manuscript needs to be improved mainly in the methodological approach.
Comments
Abstract. Needs improvement, some sentences are incomplete.
Introduction. To improve this section, data about the importance of these antimicrobial classes and the amount administrated in human medicine may be added.
Line 62-76: Confused. I suggest, first the number total of isolates and, secondly, describing the information about the first period of sample collection and after the isolates of the second period. There is repeated information in section 2.2. and 2.3.
Line 63: change “eight” to 8.
Line 67: The VITEK 2 were also used to identify the bacterial species or only for the antimicrobial susceptibility?
Methods: The antimicrobial susceptibility was performed using 15 antibiotics by MIC and 12 antibiotics by the disk diffusion method? For all isolates or only for isolates of the second period? Both methods to Antimicrobial susceptibility testing should be described in section 2.3. Add information about the used antibiotic concentration.
Results. Line 122-126 and Table 1 should be described in the Material and Methods section and removed from the results.
Line 180: change “2.1.” to 3.1.
Tables 1 and 2. The names of genes should be in italics.
Author Response
"Please see the attachment"

Round 2
Reviewer 2 Report
Comments and Suggestions for Authors
Many thanks to the authors for their response.
Before the manuscript publication, two changes are needed:
Line 116. Link do not work.
Line 200, 220: change “2.1.”, “2.2.” and “2.3.”to 3.2., 3.3. and 3.4.
